

# Crystallization and structure analysis of the core motif of the Pks13 acyltransferase domain from *Mycobacterium tuberculosis*

Mingjing Yu[1,*], Chao Dou[1,*], Yijun Gu[2] and Wei Cheng[1]

[1] Department of Respiratory and Critical Care Medicine, West China Hospital, Sichuan University, Chengdu, Sichuan, People's Republic of China
[2] National Center for Protein Science, Shanghai, People's Republic of China
* These authors contributed equally to this work.

## ABSTRACT

Type I polyketide synthase 13 (Pks13) is involved in the final step of the biosynthesis of mycolic acid in *Mycobacterium tuberculosis*. Recent articles have reported that Pks13 is an essential enzyme in the mycolic acid biosynthesis pathway, and it has been deeply studied as a drug target in Tuberculosis. We report a high-resolution structure of the acyltransferase (AT) domain of Pks13 at 2.59 Å resolution. Structural comparison with the full-length AT domain (PDB code, 3TZW, and 3TZZ) reveals a different orientation of the C-terminal helix and rearrangement of some conserved residues.

## INTRODUCTION

Tuberculosis (TB) and its drug-resistant forms are still the primary causes of mortality, surpassing other infectious diseases (*Dande & Samant, 2018*) and emphasizing the unmet clinical need for new drugs with novel mechanisms. Owing to the indispensable and specific lipids forming the envelope of *Mycobacterium tuberculosis* (*Dubnau et al., 2000*), targeting the synthesis and transport pathways of mycolic acids has always been the main route of TB drug discovery (*Bhatt et al., 2007*; *Brennan & Nikaido, 1995*; *North, Jackson & Lee, 2014*; *Wilson et al., 2013*).

Recently, powerful evidence has verified that Pks13 is an essential enzyme in the mycolic acid biosynthesis pathway (*Gavalda et al., 2009*; *Portevin et al., 2004*), and Pks13 has been extensively studied as a drug target for TB (*Aggarwal et al., 2017*; *Thanna et al., 2016*). The type-1 polyketide synthase enzyme Pks13 consists of five domains. The medial three are mandatory polyketide synthase domains, namely, the ketoacyl synthase (KS) domain, the acetyltransferase (AT) domain and the acyl carrier protein (ACP) domain. The other ACP domain is adjacent to the KS domain, and the thioesterase (TE) domain is the C-terminal portion of Pks13. The overall Pks13 topological structure has the order ACP-KS-AT-ACP-TE (Fig. 1A).

The residue Ser[55] in the N-ACP domain has been identified as a very important active site for initializing the pathway. The sfp gene encodes phosphopantetheinyl transferase,

Corresponding author
Wei Cheng, chwlab@163.com

**A**

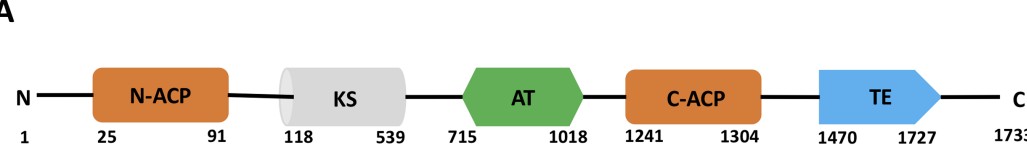

**B**

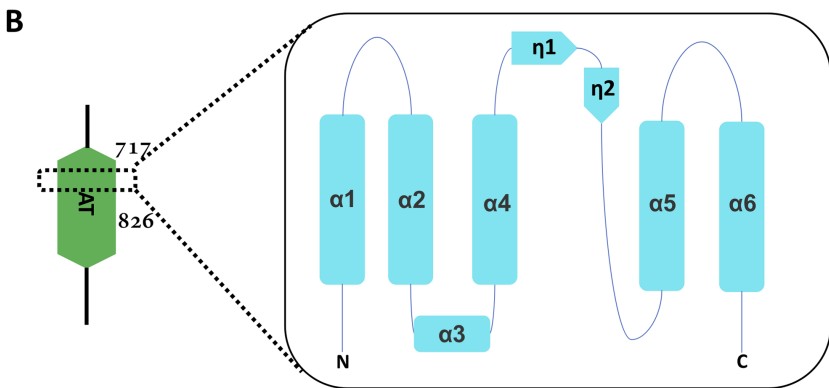

**Figure 1 The overall Pks13 domain structure has the order ACP-KS-AT-ACP-TE.** (A). The leading domain N-ACP (25–91) is colored orange. The medial three domains are mandatory PKS domains, including the KS domain (118–539), the AT domain (715–1018) and the C-ACP domain (1241–1304), colored cyan, green and orange, respectively. The TE domain (1470–1727) is located in the C terminus and colored blue. Residue numbers are given below for each domain boundary. (B). The motif resides in the AT domain ranging from Ala$^{717}$ to Arg$^{826}$. The whole topographic structure is composed of six α helixes and two short η turns, in the order of N terminus-α1-α2-α3-α4-η1-η2-α5-α6-C terminus.

which modifies ACPs by providing a P-pant arm for the general function of carrying the substrate acyl chain via a thioester bond involving its terminal thiol group (*Chalut et al., 2006*; *Gavalda et al., 2009*; *Wilson et al., 2013*). The meromycoloyl chain on the N-ACP domain is transferred to the KS domain, and the intermediate product α-alkyl β-ketothioester is produced by a Claisen-type condensation reaction with another substrate, the carboxyacyl-CoA loaded by the AT domain. The mycolic acid precursor generated by the C-terminal ACP domain is then released by the TE domain (*Abrahams & Besra, 2016*; *Dubey, Sirakova & Kolattukudy, 2002*).

Despite increasing insights into the mechanism of Pks13, no full-length structural information has been reported, except that the structures of a few domains belonging to Pks13 have been solved (*Bergeret et al., 2012*; *Herbst et al., 2016*).

Here, we report a high-resolution structure of the core motif of the AT domain. First, the full-length Pks13 protein was successfully purified, and an extended crystal screening was performed, in which the initial crystal was obtained. While attempting to phase the diffraction data of the crystal, we found that the crystallized protein suggested a degraded fragment. Then, the crystals were solved, and the N-terminal sequence was identified by mass spectrometry, the results of which were in line with the phase presented by the Se-Met crystal dataset. These results indicated that the crystallized protein was actually proteolyzed to become a fragment (Ala$^{717}$ to Arg$^{826}$). The overall crystal structure

displayed a fold similar to the reported AT domain, excluding several conformational changes relative to the reported AT domain (Protein Data Bank codes: 3TZW, 3TZZ). The structural alignment performed by the secondary structure matching (SSM) in Coot also showed a superimposition of the core motif and the AT domain with an r.m.s.d. of 1.33 Å, which was mainly attributed to the rearrangement of residues $Ala^{796}$–$Ser^{801}$. In addition, the position of residue $Ser^{801}$ that is reported to be the catalytic residue was shifted away from the active site (*Bergeret et al., 2012*; *Gavalda et al., 2009*). Furthermore, a highly conserved arginine residue, $Arg^{826}$, lost a hydrogen bond with the side chain of $Gln^{773}$, as observed in our structure. These features might all contribute to the unique state that survived proteolysis.

We believe that comprehensive structural studies of Pks13 will pave the way for structure-based antimycobacterial drug design and drug screening.

## MATERIALS AND METHODS

### Cloning, over-expression, and purification

The codon-optimized gene encoding the full-length Pks13 protein originating from *M. tuberculosis* was ligated into the *Nde*I and *Xho*I sites of the pET-28b expression plasmid (Novagen, Madison, WI, USA). The sfp gene, which encodes the P-pant transferase that serves as a kind of cofactor to modify $Ser^{55}$ in the N-ACP domain of Pks13, from *Bacillus subtilis* str.168 (*Chalut et al., 2006*) was also ligated into the *Nde*I and *Xho*I sites of the pET-21b expression plasmid (Novagen, Madison, WI, USA), and a terminator codon was added to the C-terminal end. The detailed information on these constructs is shown in Table 1. All constructed plasmids were verified by sequencing.

The constructed plasmid Pks13-pET-28b was cotransformed with sfp-pET-21b into *E. coli* strain BL21 (DE3). The bacteria containing these recombinant plasmids were grown at 310 K in M9 medium (6 g/L $Na_2HPO_4$, 3 g/L $KH_2PO_4$, 1 g/L $NH_4Cl$, 0.5 g/L NaCl, and 0.4% glucose) supplemented with 0.05 g/L kanamycin and 0.1 g/L ampicillin. When the OD600 reached 0.5, the medium was supplemented with amino acids (0.1 g/L L-lysine, L-phenylalanine, and L-threonine; 0.05 g/L L-isoleucine, L-leucine, and L-valine; and 0.1 g/L L-Se-methionine). In addition, the protein was overexpressed after the addition of 0.3 mM IPTG at 289 K for approximately 16 h. Cell pellets were harvested by 4,000 rpm centrifugation for 10 min and suspended in a solution of 1 mM PMSF, 150 mM NaCl, and 25 mM Tris/HCl (pH 8.0) suspension buffer. After sonication, we clarified the cell lysate by centrifugation at 15,000$g$ for 30 min. The supernatant containing the modified protein was applied to a nickel-affinity column (Ni-NTA; GE Healthcare, Little Chalfont, UK) preequilibrated with suspension buffer.

The resin was gradient washed with ice-cold washing buffer (25 mM Tris/HCl (pH 8.0) and 150 mM NaCl) containing 20, 30, and 40 mM imidazole, and the proteins were eluted with elution buffer (25 mM Tris/HCl pH 8.0, 150 mM NaCl, and 250 mM imidazole). Before loading onto an anion exchange column (Source Q; GE Healthcare), the eluate with 250 mM imidazole was diluted by half with buffer A (25 mM Tris/HCl (pH 8.0) and 3 mM DTT). Subsequently, the peak fractions were collected for further purification by size-exclusion chromatography (Superdex 200 10/300; GE Healthcare) in 10 mM
**Table 1 Macromolecule production information.**

| Source organism | *Mycobacterium tuberculosis*(H37Rv) | *Bacillus subtilis str.168* |
|---|---|---|
| DNA source | Full-length Pks13 | Sfp (P-pant transferase) |
| Forward primer | 5-ggaattccatatgatggcagatgtggccg-3 | 5-ggaattccatatgaagatttacggaa-3 |
| Reverse primer | 5-ccgctcgagctgtttaccaacctcg-3 | 5-ccgctcgagtcaagcggaagcgata-3 |
| Cloning vector | pET-28b | pET-21b |
| Expression vector | pET-28b | pET-21b |
| Expression host | *E. coli* strain(DE3) | *E. coli* strain(DE3) |
| Complete amino acid sequence of the construct produced | MADVAESQENAPAERA . . . . . . IEADRTSEVGKQLE | MKIYGIYMDRPLSQEENERFMSFISPEKREKCR . . . . . . PGYKMAVCAAHPDFPEDITMVSYEELL |

Tris/HCl (pH 8.0) buffer containing 100 mM NaCl. The purity of the protein was determined by 12% SDS-PAGE gels stained by Coomassie brilliant blue. The eluted protein was concentrated by a 10 kDa centrifugal filter and flash-frozen in liquid nitrogen for crystallization.

## Crystallization

The protein encoded by the constructed plasmid and labeled with Se-Met was concentrated to 12 mg/ml. Index (Hampton Research, Aliso Viejo, CA, USA) and PEG/ION (Hampton Research, Aliso Viejo, CA, USA) kits were used for the initial crystallization trials at 293 K by the sitting-drop vapor-diffusion method (*Luft & Detitta, 1995*). Each drop contained 1 µL of protein solution and an equal volume of reservoir solution.

The initial crystal was obtained from a solution of 300 mM KAc, pH 8.1, and 20% PEG 3,350. Further crystal optimization experiments were performed by systematic variation of the precipitant concentration. Ultimately, the best crystals were screened in a solution consisting of 300 mM KAc, pH 8.1, and 25% PEG 3,350. The crystals grew to full size in 10 days and were flash-frozen in liquid nitrogen with 10% glycerol added as a cryoprotectant before X-ray diffraction.

## Data collection

X-ray diffraction data were collected at 100 K using a Pilatus3 6M detector. All the datasets were obtained at beamline BL19U1 of the Synchrotron Radiation Facility in Shanghai (*Wang et al., 2016*). A total of 360 images were recorded with 0.5 s exposure at a crystal-to-detector distance of 450 mm, and a total rotation range of 360° was covered using 1.0 oscillation.

## Protein N-terminal sequence based on mass spectrometry

Regarding the dataset of the crystalized Pks13, the initial trial did not seem to provide a structure with all of the residues because of the insufficient density for many residues. After X-ray diffraction, the crystals were collected together and analyzed with SDS-PAGE gels stained by Coomassie brilliant blue. The gel with a single low molecular line was processed with the standard in-gel digestion for mass spectrometric

characterization to identify the actual location of the degraded fragment in Pks13 (*Shevchenko et al., 2006*).

### Data refinement

All datasets were processed by HKL-2000 (*Brodersen et al., 2006*). The crystal structure of the motif was solved by single-wavelength anomalous dispersion (SAD) phasing using the anomalous data collected from the Se-Met crystal. The final model was manually built in Coot (*Emsley et al., 2010*) and refined in PHENIX (*Adams et al., 2010*). The final models were validated by MolProbity and deposited in the Protein Data Bank (PDB code 5XUO).

## RESULTS

### Purification and crystallization of Pks13

The full-length Pks13 protein was successfully overexpressed in *E. coli* BL21 (DE3), and the initial crystal condition (300 mM KAc, pH 8.1, and 20% PEG 3,350) was screened. The mature lump-like crystals were optimized after a series of crystal optimization experiments, including crystallization with different detergents and additives.

### Data collection

X-ray diffraction datasets for the Se-Met-labeled crystals were obtained at beamline BL19U1 of the Synchrotron Radiation Facility in Shanghai with a wavelength of 0.97852 Å. Diffraction images for the crystals were processed using HKL-2000.

### Protein N-terminal sequence

The prepared gel was digested by trypsin, and the digestion was purified into freeze-dried peptide powder. Then, the peptide was resolved by an Orbitrap Elite LC-MS/MS for analysis. The sequenced peptides were blasted within the full-length Pks13 protein, and the crystallized fragment protein was located in the range from $Ala^{717}$ to $Arg^{826}$ (Table 2).

### Data refinement

The crystal belonged to the space group R32, with asymmetric unit cell parameters of $a = 93.694$, $b = 93.694$, $c = 97.908$, $\alpha = \beta = 90$, and $\gamma = 120$. Additionally, the phases were determined by the SAD method. The final model was manually built in Coot and refined in PHENIX to an $R_{\text{free}}$ of 26.05% with good stereochemistry. The collected and processed data are presented in Table 3.

### Overall architecture and Superimposition with AT domain

The overall structure of the core motif contains a long $\alpha$ helix, five short $\alpha$ helixes and two short $\eta$ turns, in the order of $\alpha1$-$\alpha2$-$\alpha3$-$\alpha4$-$\eta1$-$\eta2$-$\alpha5$-$\alpha6$, which constitutes a compact motif (Fig. 1B). The long $\alpha$ helix, $\alpha4$, distributes in the middle and is surrounded by the other five short $\alpha$ helixes and two short $\eta$ turns (Fig. 2A). Superimposition with the reported structure of the AT domain (PDB code 3TZZ) (*Bergeret et al., 2012*) suggested that the core motif was located in the central region of the AT domain (Fig. 2B).

**Table 2 Mass spectrum based on protein N-terminal sequencing PEP, Posterior Error Probability of the identification.**

| Sequence | Length | Mass | Charges | PEP | Score |
|---|---|---|---|---|---|
| AGFGAQHR | 8 | 842.9 | 2;3 | 0 | 201.07 |
| AGFGAQHRK | 9 | 971.07 | 3 | 0.020363 | 8.308 |
| HHGAKPAAVIGQSLGEAASAYFAGGLSLR | 29 | 2835.478 | 2;3;4;5 | 2.88E-48 | 83.418 |
| HHGAKPAAVIGQSLGEAASAYFAGGLSLRDATR | 33 | 3278.6909 | 3;4;5 | 0.012023 | 14.817 |
| KMGKSLYLR | 9 | 1094.627 | 2;3 | 0.0011449 | 41.427 |
| MGKSLYLR | 8 | 966.53207 | 2 | 5.16E-09 | 75.109 |
| MGKSLYLRNEVFAAWIEK | 18 | 2154.1296 | 3;4 | 1.24E-11 | 40.856 |
| NEVFAAWIEK | 10 | 1205.6081 | 2 | 8.84E-19 | 89.08 |
| PAAVIGQSLGEAASAYFAGGLSLR | 24 | 2305.2066 | 2;3 | 1.27E-303 | 269.26 |
| PAAVIGQSLGEAASAYFAGGLSLRDATR | 28 | 2748.4195 | 3 | 6.29E-159 | 160 |
| SLYLRNEVFAAWIEK | 15 | 1837.9727 | 2;3 | 7.27E-68 | 127.02 |
| SSGLVPR | 7 | 714.40244 | 2 | 3.61E-08 | 74.191 |

**Note:**
This value essentially operates as a *p*-value, where smaller is better.

**Table 3 X-ray data collection and refinement statistics.**

| Data set | Core motif of AT domain |
|---|---|
| Data collection | |
| X-ray source | SSRF BEAMLINE BL19U1 |
| Space group | R32 |
| Wavelength (Å) | 0.97852 |
| Resolution range (Å) | 50–2.587 |
| Total no. of reflections | 90,229 |
| No. of unique reflections | 5,358(524) |
| Completeness (%) | 99.70 |
| Redundancy | 16.999 |
| $R_{r.i.m.}$ | 0.026 |
| $I/\sigma(I)$ | 41.27(2.10) |
| Refinement | |
| Resolution range | 50–2.59 |
| Reflections: working/test | 5,083/276 |
| Final $R_{cryst}$ | 23.69% |
| Final $R_{free}$ | 26.05% |
| Rotamer outliers | 2.4% |
| Ramachandran plot: | |
| Favored/allowed/outliers (%) | 93.58/4.59/1.83 |
| Rmsd bonds(Å) | 0.003 |
| Rmsd angles (°) | 0.540 |
| PDB accession code | 5XUO |

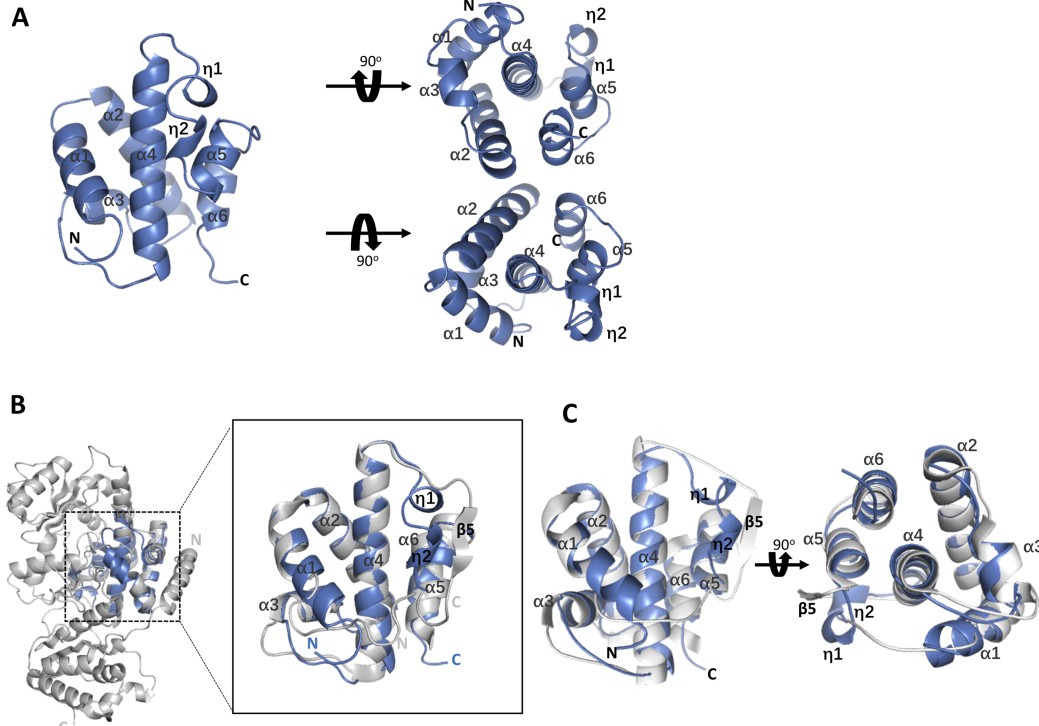

**Figure 2 The architecture of the core motif from the AT domain and structural comparison with the AT domain (PDB code 3TZW and 3TZZ).** (A). The overall topological structure of the core motif contains a long α helix in the middle, five short α helixes and two short η turns around the long helix, which constitutes a compact motif. The top and bottom views with a rotation of 90° are exhibited on the right. (B). The structures of the motif and the AT domain (PDB code 3TZZ) are superimposed together and colored blue and gray, respectively. The aligned region is zoomed in for clear observation. (C). The superimposition of the motif and the region of the AT domain that could be aligned is shown in two orthogonal views. The secondary elements are labeled in the picture.

The crystallized core motif ranging from Leu[717] to Arg[826] represents approximately one-third of the AT domain, and the overall crystal structure displays a fold similar to the reported AT domain (Fig. 2C).

Although sequence alignment showed 100% identity between the core motif and the AT domain, the secondary structure elements presented a slight conformational change from residues Ala[796] to Ser[801], for which refinement indicated two η turns instead of the β strand highlighted by red dashed square line (Fig. 3A). According to the structure of the AT domain reported by *Bergeret et al. (2012)* there was a parallel six-stranded β-sheet (β13-β12-β4-β5-β10-β11) along with the active site in the reported AT domain, while only the central β strand, β5, was presented in the motif structure and was refined as a completely different secondary element (Figs. 3B and 3C). Previous studies suggested that the conserved Ser[801] and Arg[826] could serve as a catalytic residue and binding site, respectively. The active site Ser[801] of the AT domain is located in the nucleophilic elbow between β5 and helix α10 and could directly contact the lipid substrate. Additionally, the active cite constituted the part of the highly conserved consensus sequence Gly-X-Ser-X-Gly that stabilizes the β5 strand shape (*Bergeret et al., 2012*;

# A

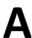

**Figure 3 Structural and sequence alignments.** (A). Structure-based sequence alignment with the whole AT domain. The secondary structural elements of the motif are given along the top of the alignment; the secondary elements of the AT domain (Protein Data Bank codes 3TZZ and 3TZW) are shown below. The difference between the structures is circled by the red dashed line. (B). The six β-sheets (β13-β12-β4-β5-β10-β11) (Protein Data Bank codes 3TZW and 3TZZ) are presented in the figure, and β5 is aligned with the η turns of the motif. (C). Some conserved residues show a totally different topographic structure, and residues Lys[793], Pro[794], Ala[795], Ala[796], Val[797], Ile[798], Gly[799], Gln[800], and Ser[801] are shown as sticks.

*Serre et al., 1995*). In our work, the topographic conformation of Ala[796] to Ser[801] was transformed into two relatively disordered η turns, along with the conformational change of the position of Ser[801] dislocating from the substrate. Furthermore, the side chain of the binding site Arg[826] was also stretched to the reverse side of α5 and lost its interaction with Gln[773]. However, this side chain formed direct hydrogen bonds with the negative side chain of the lipid substrate, and the conformation was held in position

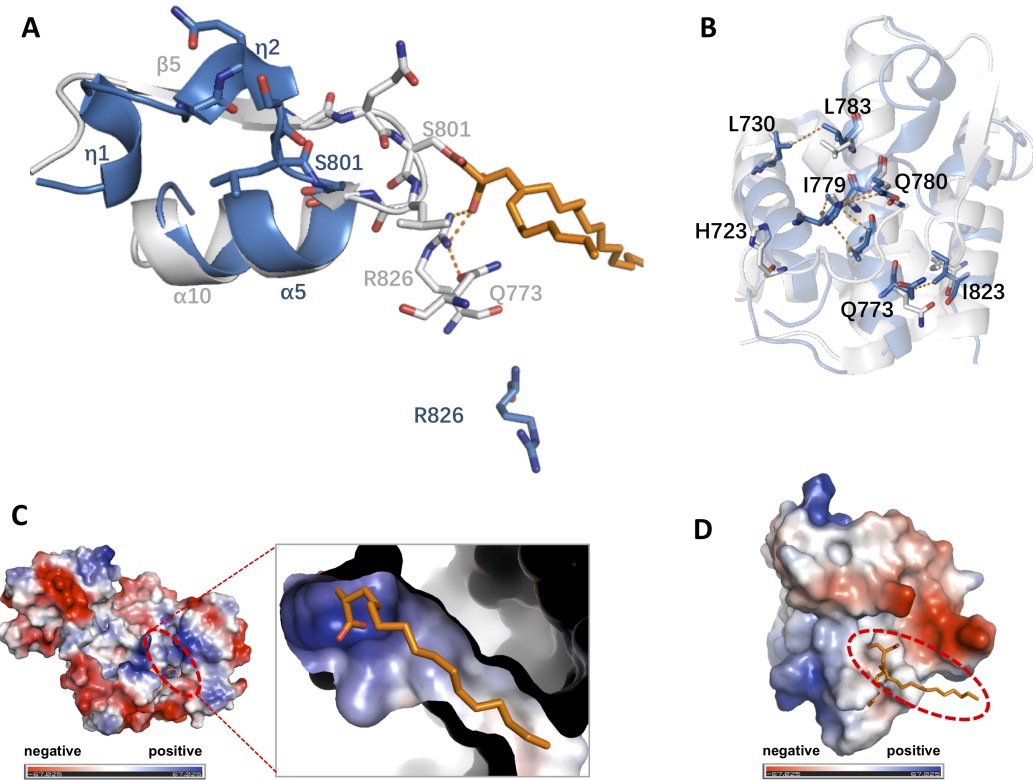

**Figure 4 Structural comparison between different states of AT domain.** (A). Detailed description of the active site of the motif compared with the AT domain. The nucleophilic elbow comprising strand β5 and helix α10 in the AT domain corresponding to helix α5 and two helical η turns, respectively, is shown in cartoon representation. Important residues defining the active site are shown and labeled. Hydrogen bonds are represented by orange dotted lines. The lipid substrate colored orange is shown as sticks. (B). Some apolar contacts among α1 (His$^{723}$, Leu$^{730}$) and the long α4 (Gln$^{773}$, Ile$^{779}$, Gln$^{780}$, Leu$^{783}$) and α5 (Ile$^{823}$) residues are shown and labeled. (C). Electrostatics calculations for the AT domain (Protein Data Bank code 3TZZ) revealed the presence of an electropositive area corresponding to the floor of the active site cavity bound with a lipid substrate. The surface representation was generated by PyMOL and colored according to its electrostatic potential (positive potential, blue; negative potential, red). The substrate cavity was highlighted by a dotted red circle and zoomed at the right panel. (D). Electrostatics calculations for the motif in this work revealed the electrostatic potential transformation from an electropositive state to an electronegative state. The substrate originating from the AT domain (Protein Data Bank code 3TZZ) was docked on the floor of the catalytic cavity and was highlighted with a red dotted circle.

through a strong hydrogen bond interaction with the side chain of Gln$^{773}$ in the AT domain (Fig. 4A).

The structural alignment performed by SSM in Coot (*Emsley & Cowtan, 2004*) also showed that the superimposition of the core motif and the AT domain had an r.m.s.d. of 1.33 Å; along with the conformational changes, this alignment might also suggest a more compact crystal packing state than that of the AT domain. According to a close view of the superimposition, some particular apolar contacts among α1 (His$^{723}$, Leu$^{730}$) and the long α4 (Gln$^{773}$, Ile$^{779}$, Gln$^{780}$, and Leu$^{783}$) and α5 (Ile$^{823}$) residues all contribute to the stabilization of the unique state (Fig. 4B). Electrostatic calculations of the AT domain (Protein Data Bank code 3TZZ) revealed the presence of an electropositive area
corresponding to the floor of the active site cavity due to the presence of Ser$^{801}$ and Arg$^{826}$ (Fig. 4C). Comparison of the electrostatic potential surface presentation of the motif indicated that the surface of the active site cavity was transformed to an electronegative state (Fig. 4D).

## DISCUSSION

The synthesis and transport pathways of mycolic acids in *M. tuberculosis* have always been a critical drug target. These mycolic acids serve as the primary defense to counteract the low permeability of the envelop to many hydrophilic molecules. Many biochemical and structural studies have sought to elucidate the participation of Pks13 in the synthesis of the lipid complex. Obtaining the structure of Pks13 is of great significance in drug screening, as many inhibitors have been reported to target Pks13 or its individual domains.

The structure of the fragment of the AT domain provides a relatively new perspective of a unique state that can evade proteolysis. We have determined the 2.59 Å high-resolution crystal structure of a partial AT domain from the *M. tuberculosis* Pks13 protein. The overall structure of the core motif of the AT domain is similar to the corresponding part of the reported AT domain, with slight conformational differences. Some conserved residues showed a completely different secondary structure. Residues Ala$^{796}$, Val$^{797}$, Ile$^{798}$, Gly$^{799}$, Gln$^{800}$, and Ser$^{801}$ formed a β strand in the previously reported AT domain (PDB codes 3TZW, and 3TZZ), which instead refined as a flexible loop conformation in the motif structure. In contrast to the typical structure of the whole AT domain containing a palm-shaped parallel six-stranded β sheet, in which β5 is located in the middle of a connection with the other five β strands. In our work, the β-sheet structure was disrupted along with loosing connections among these β strands due to the conformational changes. Actually, there was less possibility of the AT domain remaining the same because of the conformational changes from Ala$^{796}$ to Ser$^{801}$, which tend to confirm the speculation that the conformational changes are a tactic to evade proteolysis. With the structural alignment performed by SSM in Coot, the superimposition of the core motif and the AT domain shows an r.m.s.d. of 1.33 Å. The novel packed structure formed by these bundles seems tighter than the AT domain, which is especially reflected in the apolar contacts among α1 (His$^{723}$, Leu$^{730}$) and the long α4 (Gln$^{773}$, Ile$^{779}$, Gln$^{780}$, and Leu$^{783}$) and α5 (Ile$^{823}$) residues. These apolar contacts among the residues might strengthen the interactions of α4 with other helixes to form a more stable packing state.

Additionally, the active site Ser$^{801}$, which plays a critical role in catalytic activity, was dislocated away from the substrate cavity to the inner position of the core motif. The nucleophilic elbow of α10 and β5 also transformed from an electropositive state to an electronegative state which indicates an unsuitable state to absorb a substrate. In summary, the conformational change of residues from Ala$^{796}$ to Ser$^{801}$ and the rearrangement of residues Gln$^{773}$, Ser$^{801}$, and Arg$^{826}$ might all suggest that the degraded fragment formed a unique crystal packing state to survive proteolysis. In other words, the fragment forms a relatively stable state in contrast to the AT domain in such conditions. This work might provide new insight into the core motif of the AT domain. Our work also provides a structural basis for protein engineering.

However, the overall structure of Pks13 is still unrevealed, and its mechanism is yet unknown. More work should be performed, and we hope that our present work will provide some assistance.

## ACKNOWLEDGEMENTS

We gratefully acknowledge the assistance of the staff of BL19U1 at the Shanghai Synchrotron Radiation Facility (SSRF) for their assistance with X-ray diffraction data collection.

### Funding

This work was supported by the National Natural Science Foundation of China (No. 00402354011009). The funders had no role in study design, data collection and analysis, decision to publish, or preparation of the manuscript.

### Grant Disclosures

The following grant information was disclosed by the authors:
National Natural Science Foundation of China: 00402354011009.

### Competing Interests

The authors declare that they have no competing interests.

### Author Contributions

- Mingjing Yu performed the experiments, contributed reagents/materials/analysis tools, prepared figures and/or tables, authored or reviewed drafts of the paper, approved the final draft.
- Chao Dou performed the experiments, contributed reagents/materials/analysis tools, prepared figures and/or tables, authored or reviewed drafts of the paper, approved the final draft.
- Yijun Gu analyzed the data, authored or reviewed drafts of the paper, approved the final draft.
- Wei Cheng conceived and designed the experiments, authored or reviewed drafts of the paper, approved the final draft.

### Data Availability

The coordinates and structure factors are deposited at RCSB PDB, and the assigned PDB code is 5XUO: http://www.rcsb.org/pdb/home/home.do.

### Supplemental Information

Supplemental information for this article can be found online at http://dx.doi.org/10.7717/peerj.4728#supplemental-information.

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
