# Peer review of "Crystallization and structure analysis of the core motif of the Pks13 acyltransferase domain from Mycobacterium tuberculosis"

_PeerJ, doi:10.7717/peerj.4728_

## Round 0.1 · original submission · Major Revisions

Dear Mingjing and colleagues:

I am sorry for the long delay with the review of your manuscript. We struggled with finding reviewers. I know have received three reviews, and there are many concerned raised by all three reviewers. Please read over these reviews, and consider revising your work for resubmission. I do think the work required to improve your manuscript is substantial, yet if addressed, will likely result in more favorable review.

Good luck,
-joe

Reviewer 1 ·

Basic reporting

no comment

Experimental design

no comment

Validity of the findings

no comment

Additional comments

Yu et al report the purification of recombinantly expressed, full length Pks13 from Mycobacterium tuberculosis (MtPks13), together with the crystallization and structure determination of a 110-residue proteolytic fragment. The 110-residues are part of the acyltransferase domain in the polyketide synthases, that is responsible for substrate selection and transfer to a dedicated acyl carrier protein. The structure does not provide significant new information, as the crystal structure of the intact 52kDa MtPks13-AT domain is deposited in the PDB (PDB ID: 3TZW.pdb) and well described by Bergeret et al (2012) (doi: 10.1074/jbc.M111.325639). The authors should include detailed comparison to 3TZW.pdb, e.g. rmsd values (http://www.ebi.ac.uk/msd-srv/ssm/cgi-bin/ssmserver). The reported minor structural differences do not significantly support the authors conclusion that this structure represents an inactivated state of the enzyme, which would have been useful for drug discovery.

The manuscript would benefit from language proofreading from a native speaking colleague. In the crystallographic table are some points that require attention:
Sodium is stated as a ligand, but the submitted PDB file, does not contain a Sodium ion.
No rotamer outliers are stated, but the provided PDB-File has 2.4% rotamer outliers (according to wwwPDBvalidate)
The clashscore according to wwwPDBvalidate is 11, higher then current standards.

Reviewer 2 ·

Basic reporting

no comment

Experimental design

no comment

Validity of the findings

no comment

Additional comments

Yu et al. report the structure of the core motif of the acyltransferase domain of PKS13 from Mycobacterium tubercolosis. PKS13 is an important drug target to inhibt the cell wall biosynthesis of the bacterium.

Although the structural data seems solid the manuscript needs intensive improvement for publication. Some aspects are not clear (or clear enough). As an example, the strutcure reported here seems to be a degradation product of the full-lenght protein production. This is stated in lines 43-45 but it remains unclear if this is observed during protein purification or if degradation is occuring during the crystallization process? Another example: Why was Se-Met-labeled protein used although an MR model is available? The stucture comparison to the full length AT domain lacks important information like the rmsd. Additionally, an rmsd polt (rmsd vs residue number) might helpful.

The language of the manuscript serious corrections, e.g.

line 30:
.. locate in the C-terminal."

line 59:
...the C-terminal had a terminator codon."

line 111:
cell parameter units missing

line 120:
"In our structure..."

...

The research question of the article aims the desciption of different conformational states of the AT core domain. The structure reported here shows conformational changes in comparison to the full-length AT domain (pdb-code 3tzz) but it remains completely unclear, if these changes are really a different state of the protein, or just a result of protein degradation or just an artifact of the crystallization of a core protein. Further experiments and crystal packing analysis are needed. The manuscript totally ignores crystal packing influences.

The validity of the finding is my major concern of the manuscript in its current state. The authors claim that the conformational changes might indicate a different state of the enzyme. But from my point of view this is not experimentally confirmed.

·

Basic reporting

This manuscript describes the 2.6 Å crystal structure of the N-terminal region of the Pks13 acyltransferase (AT) domain from M. tuberculosis. More extensive referencing needs to be provided, for example, concerning the description of catalytic mechanism of the condensing enzyme Pks13 and the work on phosphopantetheinyl (P-pant) transferase characterization and function in Mtb. The major focus of the paper is on the conformational changes occurring at the C-terminal extremity of this fragment with a possible link to the activation state of the enzyme. The crystallized fragment represents approximately one third of the AT domain, for which a structure has been published by Bergeret and colleagues (JBC 2012).

Experimental design

All the classical methods leading to structure determination have been described in details. The insert was cloned into a pET28 vector, transformed into E. coli BL21(DE3) cells, purified following a three-steps procedure, and then crystallized.

Major points
1- There is no information on how this protein fragment has been selected.
2- The use of the 4’-phosphopantetheinyl transferase Sfp from B. subtilis is not justified as modification occurs onto the ACP domain and not onto the AT domain, the subject of this work. Please explain
3- The authors should justify the use of selenomethionine labeled protein considering that the X-ray structure of the full-length AT domain has been published (PDB code 3TZZ).

Validity of the findings

The Pks13 AT domain extends from amino acid 712 to amino acid 1035. The X-ray structure of a Pks13 fragment encompassing the full-length AT domain (amino-acid positions 597 to 1059) has previously been described (Bergeret et al. JBC 2012). Here, the crystallized fragment starts at position 717 and ends at position 826, in the middle of helix α11.

Major point
One structural feature of the full-length AT domain is the presence of a six-stranded β-sheet (β13-β12-β4-β5-β10-β11). Only residues forming the central β-strand β5 are present in the actual structure, at the C-terminal end. It is thus hard to imagine that the conformational changes observed in this region reflect a change in the activation state of the enzyme rather than a structural artefact arising from the absence of neighboring β-strands. In the absence of any experimental validation, this is pure speculation in my view.

Minor points
1- Line 22 – Wrong reference
2- Line 33 – Sfp from B. subtilis. Mtb also has two PPTases named AcpS and PptT.
3- Line 48 – It seems premature to speculate on the orientation of the C-terminal helix considering that it is truncated.
4- Line 96 and 111 – Missing ° and/or Å
5- Line 111 and table 2 – Space group R32:H ?
6- Reference section with numbering issues
7- Figure 1 – Missing information on amino-acid positions, schematic representation in the legend
8- Figure 2 – A structure-based sequence alignment with 3TZZ would be greatly appreciated. Also, indicate secondary structure elements on both sequence alignment and structure representation.
9- Figure 3 – Legend and figures need to be modified. The term “channel-like cave” is not appropriate. A surface representation may be useful. A close-up view of figure 3b is necessary.

---

## Round 0.2 · Major Revisions

Dear Mingjing and Wei,

I have sent your revision to the original reviewers, and upon receipt of their second reviews, it is clear that there remains a lack of novelty with your study. I am sorry I cannot be more positive at this time, but I am rejecting your manuscript, although with encouragement to resubmit. I have thus marked the paper “major revision” with an understanding that you will revise the work according to the reviewer’s suggestions (and mine). In your revision, please address the impossibility of the conformational change happening in the full AT domain (and therefore most likely not having anything to do with the presence/absence of ligand elsewhere). The differences in structure between this fragment and the AT domain are most likely spurious: the new conformation of the 793-803 segment is completely impossible in the full-length domain because it would fall on top of the 711-718 beta-strand and disrupt the whole beta-sheet. Thus, the current focus of the paper is inappropriate, and its speculations regarding the significance/origin of those changes are also inappropriate. You might consider deviating from this focus (stability change upon bonding/unbinding substrate as an explanation of the 793-803 segment conformation change) and just emphasizing the stability of the core motif that "survived" proteolysis. Such a change in focus likely would create a dialogue that is much more meaningful to the literature. Finally, the manuscript still has serious linguistic problems and definitely should be carefully edited at the same time.

Thanks again for considering PeerJ for publication of your work, and should you choose to resubmit your work, I look forward to seeing the revision.

Sincerely,

Joe

Reviewer 1 ·

Basic reporting

The authors have answered all questions adequately. The manuscript can be accepted.

Experimental design

-

Validity of the findings

-

Additional comments

-

·

Basic reporting

no comment

Experimental design

no comment

Validity of the findings

no comment

Additional comments

1- Basic reporting
The authors have made some significant changes to the paper, including new figures, references. Unfortunately, this work does not provide significant new information considering that (i) no functional data have been provided and (ii) the intact full-length AT domain X-ray structure has previously been described in great details by Bergeret and colleagues.

Some important facts…
1- Writing problems throughout: there are myriad writing problems including diction (word use, lack of prepositions and plurals, etc) and grammatical inaccuracies.... It would make sense to have a professional or at least a native English speaking scientist review the manuscript and correct these errors. There are just too many.
2- Line 34 “Phosphopantetheinyl (P-pant) transferase (PPtT), encoded by the sfp gene…” should read Phosphopantetheinyl transferase (PPTase)
3- Line 59 “The sfp gene from Bacillus subtilis str.168 (Chalut et al., 2006), which encodes the 4 -phosphopantetheinyl (P-pant) transferase PptT…” sfp and pptT are two different genes.
4- Table 1 - Sfp (PptT). Same as above.

---

## Round 0.3 · accepted · Accept

Dear Mingjing and Wei,

Thanks for making the recommended edits and congratulations on acceptance! It was a long process, but in the end you prevailed!

Best,

-joe

#